# Hindsight Task Relabelling:
# Experience Replay for Sparse Reward Meta-RL

**Charles Packer**
UC Berkeley

**Pieter Abbeel**
UC Berkeley

**Joseph E. Gonzalez**
UC Berkeley

## Abstract

Meta-reinforcement learning (meta-RL) has proven to be a successful framework for leveraging experience from prior tasks to rapidly learn new related tasks, however, current meta-RL approaches struggle to learn in sparse reward environments. Although existing meta-RL algorithms can learn strategies for adapting to new sparse reward tasks, the actual adaptation strategies are learned using hand-shaped reward functions, or require simple environments where random exploration is sufficient to encounter sparse reward. In this paper, we present a formulation of hindsight relabeling for meta-RL, which relabels experience during meta-training to enable learning to learn entirely using sparse reward. We demonstrate the effectiveness of our approach on a suite of challenging sparse reward goal-reaching environments that previously required dense reward during meta-training to solve. Our approach solves these environments using the true sparse reward function, with performance comparable to training with a proxy dense reward function.

## 1 Introduction

Reinforcement learning (RL) has seen tremendous success applied to challenging games (Mnih et al., 2015; Silver et al., 2017) and robotic control (Lillicrap et al., 2015; Levine et al., 2016), driven by advances in compute and the use of deep neural networks as powerful function approximators in RL algorithms. However, agents trained using deep RL often struggle to meaningfully utilize past experience to learn new tasks, even if the new tasks differ only slightly from tasks seen during training (Zhang et al., 2018; Packer et al., 2018; Cobbe et al., 2019). In contrast, humans are adept at utilizing prior experience to rapidly acquire new skills and adapt to unseen environments.

Meta-reinforcement learning (meta-RL) aims to address this limitation by extending the RL framework to explicitly consider structured distributions of tasks (Schmidhuber, 1987; Bengio et al., 1990; Thrun & Pratt, 1998). Whereas conventional RL is concerned with learning a single task, meta-RL is concerned with *learning to learn*, that is, learning how to quickly learn a new task by leveraging prior experience on related tasks. Meta-RL methods generally utilize a limited amount of experience in a new environment to estimate a latent task embedding which conditions the policy, or to compute a policy gradient which is used to directly update the parameters of the policy.

A major challenge in both RL and meta-RL is learning with sparse rewards. When rewards are sparse or delayed, the adaptation stage in meta-RL becomes extremely difficult: inferring the task at hand requires receiving reward signal from the environment, which in the sparse reward setting only happens after successfully completing the task. Due to this inherent incompatibility between meta-RL and sparse rewards, existing meta-RL algorithms that consider the sparse reward setting either only work in simple environments that do not require temporally-extended exploration strategies for adaptation (Duan et al., 2016; Stadie et al., 2018), or train exclusively using dense reward functions (Gupta et al., 2018b; Rakelly et al., 2019), which are designed to encourage an agent to learn adaptation strategies that can be directly applied to the original sparse reward setting.

35th Conference on Neural Information Processing Systems (NeurIPS 2021).

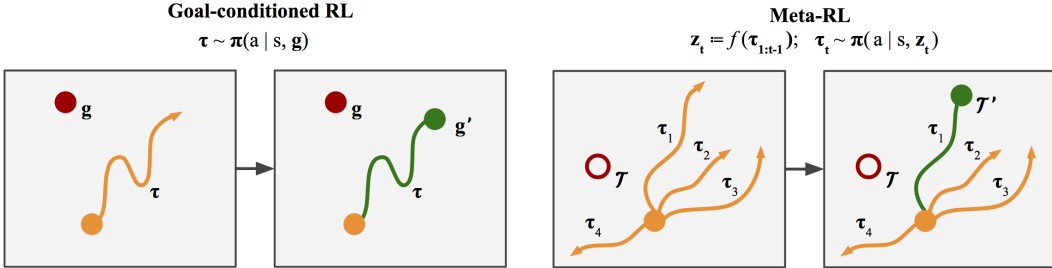

(a) Hindsight relabeling in goal-conditioned RL  (b) Hindsight Task Relabeling (HTR) in meta-RL

Figure 1: In goal-conditioned RL (a), an agent must navigate to a provided goal location $\mathbf{g}$ (filled circle, revealed to the agent). An unsuccessful attempt for goal $\mathbf{g}$ provides no sparse reward signal, but can be relabelled as a successful attempt for goal $\mathbf{g}'$, creating sparse reward that can be used to train the agent. In meta-RL (b), the task $\mathcal{T}$ (i.e., goal, hollow circle) is never revealed to the agent, and instead must be inferred using experience on prior tasks and limited experience ($\tau_{1:t-1}$) on the new task. In (b), there is no shared optimal task $\mathcal{T}'$ to relabel all attempts with. HTR relabels each attempt $\tau$ under its own hindsight task $\mathcal{T}'$, and modifies the underlying meta-RL training loop to learn adaptation strategies on the relabelled tasks. Note that we include multiple trajectories $\tau$ in (b) vs a single trajectory in (a) to highlight the adaptation stage in meta-RL, which does not exist in goal-conditioned RL and requires significantly different sampling and relabeling procedures.

In this paper, we show that the concept of *hindsight relabeling* (Andrychowicz et al., 2017) from conventional RL can be applied in meta-RL to enable learning to learn in the sparse reward setting. Our key insight is that data collected on the true training tasks can be relabeled as pseudo-expert data for easier hindsight tasks, bootstrapping meta-training with the reward signal needed to train the agent. We introduce a new algorithm for off-policy meta-RL, which we call *Hindsight Task Relabeling* (HTR), and demonstrate its effectiveness by achieving state-of-the-art performance on a collection of challenging sparse reward environments that previously required shaped reward to solve. Not only is our approach able to learn adaptation strategies only using sparse reward, but it also learns strategies with comparable performance to existing approaches that use shaped reward functions.

## 2   Related Work

In meta-RL, an agent learns an adaptation strategy by repeatedly adapting to training tasks in the inner loop of *meta-training*, with the hope that the learned adaptation procedure will generalize to new, unseen tasks during *meta-testing*. Context-based methods use recent experience (i.e., context, in the form of trajectories or transitions) in a new task to estimate a latent task embedding, and differ mainly in how this embedding is computed: Duan et al. (2016); Wang et al. (2016); Mishra et al. (2018) propose aggregating context into the hidden state of the policy, whereas Rakelly et al. (2019) propose explicitly feeding the task embedding as input to the policy. Gradient-based methods use recent context to update differentiable hyperparameters (Xu et al., 2018b), loss functions (Sung et al., 2017; Houthooft et al., 2018), or to directly update policy parameters (Finn et al., 2017; Stadie et al., 2018; Xu et al., 2018a; Zintgraf et al., 2018; Gupta et al., 2018b; Rothfuss et al., 2019).

Many of the aforementioned approaches struggle to learn effective exploration strategies for tasks with sparse or delayed rewards. Methods that directly update the policy either implicitly (e.g., with a hidden state) or explicitly (e.g., with gradients) are generally unable to learn a policy that meaningfully explores, since the primary source of randomness is action-space, and thus is time-invariant. Gupta et al. (2018b) and Rakelly et al. (2019) propose using a probabilistic task variable that is sampled once per episode, which makes the primary source of variability task-dependent and enables temporally-extended exploration. While this approach enables effective adaptation in the sparse reward setting, both Gupta et al. (2018b) and Rakelly et al. (2019) learn the actual adaptation strategies using shaped (dense) reward functions during meta-training. In the environments they consider, the adaptation strategies learned using dense rewards generalize to sparse rewards despite the difference in reward structure; however, engineering a suitable reward function can be costly (Ng et al., 1999) and error prone (Clark & Amodei, 2016), and thus it is highly desirable to devise a mechanism for learning to learn in the sparse reward setting that does not require an auxiliary dense reward function.

Our proposed method is closely related to prior work in unsupervised meta-RL and goal generation. Unsupervised meta-RL (Jabri et al., 2019; Gupta et al., 2018a) considers the meta-RL setting without access to a training task distribution; training tasks are instead self-generated with the aim of learning skills useful for the test task distribution. Our proposed method also generates a curriculum of training tasks, but with the intent of learning adaptation strategies in the sparse reward setting, as opposed to learning transferable skills in a setting with no rewards at all. Several methods exist for curriculum generation in the context of goal-conditioned policies, including adversarial goal generation (Florensa et al., 2018) and goal relabeling (Kaelbling, 1993; Andrychowicz et al., 2017; Levy et al., 2017; Nair et al., 2018). Recent work (Li et al., 2020; Eysenbach et al., 2020) has proposed the use of inverse RL to generalize goal relabeling to broader families of tasks. Unlike prior work on goal and task relabeling that use goal- or task-conditioned policies, we focus on the meta-RL setting where the task is unknown to the policy and must be inferred through deliberate and coherent exploration.

## 3 Background

In this section we formalize the meta-RL problem setting and briefly describe two algorithms which our approach, Hindsight Task Relabeling, builds on: Probabilistic Embeddings for Actor-critic RL (PEARL) and Hindsight Experience Replay (HER).

### 3.1 Meta-Reinforcement Learning (Meta-RL)

Conventional RL assumes environments are modeled by a Markov Decision Processes (MDP) $M$, defined by the tuple $(\mathcal{S}, \mathcal{A}, p, r, \gamma, \rho_0, T)$ where $\mathcal{S}$ is the set of possible states, $\mathcal{A}$ is the set of actions, $p : \mathcal{S} \times \mathcal{A} \times \mathcal{S} \to \mathbb{R}_{\geq 0}$ is the transition probability density, $r : \mathcal{S} \times \mathcal{A} \to \mathbb{R}$ is the reward function, $\gamma$ is the discount factor, $\rho_0 : \mathcal{S} \to \mathbb{R}_{\geq 0}$ is the initial state distribution at the beginning of each episode, and $T$ is the time horizon of an episode (Sutton & Barto, 2017).

Let $\mathbf{s}_t$ and $\mathbf{a}_t$ be the state and action taken at time $t$. At the beginning of each episode, $\mathbf{s}_0 \sim \rho_0(\cdot)$. Under a stochastic policy $\pi$ mapping a sequence of states to actions, $\mathbf{a}_t \sim \pi(\mathbf{a}_t \mid \mathbf{s}_t, \cdots, \mathbf{s}_0)$ and $\mathbf{s}_{t+1} \sim p(\mathbf{s}_{t+1} \mid \mathbf{s}_t, \mathbf{a}_t)$, generating a trajectory $\tau = \{\mathbf{s}_t, \mathbf{a}_t, r(\mathbf{s}_t, \mathbf{a}_t)\}$, where $t = 0, 1, \cdots$. Conventional RL algorithms, which assume the environment is fixed, learn $\pi$ to maximize the expected reward per episode $J_M(\pi) = \mathbb{E}^\pi \left[ \sum_{t=0}^T \gamma^t \mathbf{r}_t \right]$, where $\mathbf{r}_t = r(\mathbf{s}_t, \mathbf{a}_t)$. They often utilize the concepts of a value function $V_M^\pi(\mathbf{s})$, the expected reward conditional on $\mathbf{s}_0 = \mathbf{s}$ and a state-action value function $Q_M^\pi(\mathbf{s}, \mathbf{a})$, the expected reward conditional on $\mathbf{s}_0 = \mathbf{s}$ and $\mathbf{a}_0 = \mathbf{a}$.

Meta-RL algorithms assume that there is a distribution of tasks $p(\mathcal{T})$, and usually maximize the expected reward over the distribution, $\mathbb{E}_{\mathcal{T} \sim p(\mathcal{T})}^\pi [J_{\mathcal{T}}(\pi)]$. The distribution of tasks corresponds to a distribution of MDPs $M(\mathcal{T})$, where $\mathcal{T}$ can paramaterize an MDP's dynamics or reward. In this work we focus on the latter case, where each $\mathcal{T}$ implies an MDP with a different reward function under the same dynamics. The agent is trained on a set of training tasks $\mathcal{T}_{train} \sim p(\mathcal{T})$ during meta-training, and evaluated on a held-out set of testing tasks $\mathcal{T}_{test} \sim p(\mathcal{T})$ during meta-testing.

An important distinction between meta-RL and multi-task or goal-conditioned RL is that in meta-RL, the task is never explicitly revealed to the agent or policy $\pi$ through observations $\mathbf{s}$. Instead, $\pi$ must explore the environment to infer the task to maximize expected reward. This crucial difference is one reason why Hindsight Experience Replay cannot be directly applied to the meta-RL setting.

### 3.2 Off-Policy Meta-Reinforcement Learning

We demonstrate the effectiveness of our approach by building on top of Probabilistic Embeddings for Actor-critic RL (PEARL) by Rakelly et al. (2019), an off-policy meta-RL algorithm which itself is built on top of soft actor-critic (SAC) by Haarnoja et al. (2018). SAC is an actor-critic algorithm based on the maximum entropy RL objective that optimizes a stochastic policy $\pi$ with off-policy data. In addition to the policy (actor) $\pi$, SAC concurrently learns twin state-action value functions (critics) $Q$. Transitions collected using the policy $\pi$ are added to the replay buffer $B$, and $Q$ and $\pi$ are optimized using loss functions regularized by the entropy of $\pi$.

Rakelly et al. (2019) extend SAC to the meta-RL setting by adding a context encoder that uses recent history in an environment to estimate a task embedding, which is then utilized by the policy.

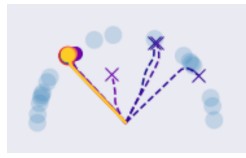 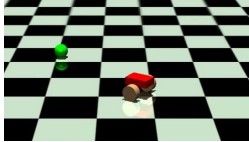 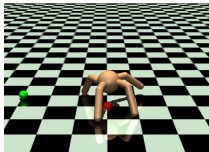

(a) Point Robot         (b) Wheeled Locomotion         (c) Ant Locomotion

Figure 2: Sparse reward environments for meta-RL that require temporally-extended exploration. In each environment, the task (the top-left circle in (a), the green sphere in (b) and (c)) is not revealed to the agent via the observation. The agent must instead infer the task through temporally-extended exploration (illustrated by the dotted lines in (a)), since no reward signal is provided until the task is successfully completed. Prior meta-RL methods such as PEARL (Rakelly et al., 2019) and MAESN (Gupta et al., 2018b) are only able to (meta-)learn meaningful adaptation strategies using dense reward functions. Our approach, Hindsight Task Relabeling (HTR), can (meta-)train with the original sparse reward function and does not require additional dense reward functions.

Specifically, PEARL uses a network $q_\phi(\mathbf{z} \mid \mathbf{c})$ that takes as input recently collected data (i.e., context $\mathbf{c}$) to infer a probabilistic latent context variable $Z$. Samples from the latent context variable condition the actor, $\pi_\theta(\mathbf{a} \mid \mathbf{s}, \mathbf{z})$, and critic, $Q_\theta(\mathbf{s}, \mathbf{a}, \mathbf{z})$. Instead of a single replay buffer (as in SAC), there are $T$ separate buffers $B_{i=1...T}$, corresponding to the number of tasks sampled for $\mathcal{T}_{train}$. The separate per-task replay buffers enable sampling off-policy (but on-*task*) context during the inner loop of meta-training, which significantly improves data efficiency.

During meta-training, transitions are collected for each $B_i$ by sampling $\mathbf{z} \sim q_\phi(\mathbf{z} \mid \mathbf{c})$ and acting according to $\pi_\theta(\mathbf{a} \mid \mathbf{s}, \mathbf{z})$. $\mathbf{z}$ is periodically sampled during data collection such that the context contains transitions collected using the policy conditioned on the prior, as well the same policy conditioned on the posterior. Gradients used to optimize $q_\phi$, $\pi_\theta$ and $Q_\theta$ are computed in task-specific batches across a random subset of training tasks $\mathcal{T}_i \sim p(\mathcal{T})$, each with an associated replay buffer $B_i$. During meta-testing, for each test task $\mathcal{T}_i \sim p(\mathcal{T})$, an empty buffer $B_i$ is initialized, $\mathbf{z}$ is sampled from the prior, and the policy conditioned on $\mathbf{z}$ is rolled out. As additional context is added to the replay buffer, the belief over $\mathbf{z}$ narrows, enabling the policy to adapt to the task at hand.

### 3.3 Hindsight Experience Replay

Hindsight Experience Replay (HER) (Andrychowicz et al., 2017) is a method for off-policy goal-conditioned RL in environments with sparse reward. The key insight behind HER is that unsuccessful attempts (i.e., attempts that received no reward signal) generated via a goal-conditioned policy can be relabelled as successful attempts for a 'hindsight' goal that was actually achieved, e.g., the final state (see Figure 1). HER can be applied to any off-policy RL algorithm that uses goal-conditioned policies (where the actor and/or critic receive the desired goal as part of the state, i.e., $\pi(\mathbf{a} \mid \mathbf{s}, \mathbf{g})$ and $Q(\mathbf{s}, \mathbf{a}, \mathbf{g})$). When a transition $\{\mathbf{s}_t, \mathbf{a}_t, r(\mathbf{s}_t, \mathbf{a}_t, \mathbf{g}) \mid \mathbf{g}\}$ is sampled from the replay buffer during training, with some probability HER rewrites the desired goal $\mathbf{g}$ with an achieved goal $\mathbf{g}'$ and recomputes the reward under the new goal. The modified transition $\{\mathbf{s}_t, \mathbf{a}_t, r(\mathbf{s}_t, \mathbf{a}_t, \mathbf{g}') \mid \mathbf{g}'\}$ is used to optimize $\pi(\mathbf{a} \mid \mathbf{s}, \mathbf{g}')$ and $Q(\mathbf{s}, \mathbf{a}, \mathbf{g}')$. Hindsight relabeling generates an implicit curriculum: initially, sampled transitions are rewritten with relatively 'easy' goals achievable with random policies, and as training progresses, relabelled goals are increasingly likely to be near the true desired goals.

## 4 Leveraging Hindsight in Meta-Reinforcement Learning

Our algorithm, Hindsight Task Relabeling, applies ideas from hindsight relabeling to the meta-RL setting to enable learning adaptation strategies for tasks with sparse or delayed rewards. Similar to how policies in HER are conditioned on a goal $\mathbf{g}$, policies in context-based meta-RL are conditioned on a latent task embedding $\mathbf{z}$. A crucial difference in meta-RL is that the task is hidden from the agent: the agent must infer relevant features of the task at hand using experience on prior tasks and a limited amount of experience gathered in the environment. Our key insight is that an unsuccessful experience collected in an *unknown* task can be relabelled as a successful experience for a *known* hindsight task, i.e., a transition $\{\mathbf{s}_t, \mathbf{a}_t, r(\mathbf{s}_t, \mathbf{a}_t, \mathcal{T})\}$ generated under an unknown $\mathcal{T}$ can be rewritten under a known $\mathcal{T}'$ as $\{\mathbf{s}_t, \mathbf{a}_t, r(\mathbf{s}_t, \mathbf{a}_t, \mathcal{T}')\}$, regardless of whether $\mathcal{T}' \in \mathcal{T}_{train}$.

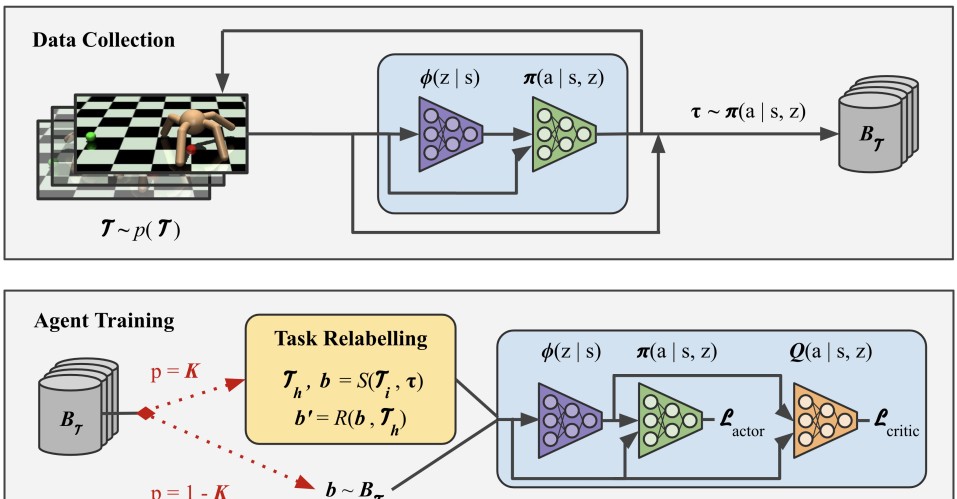

Figure 3: Illustration of Hindsight Task Relabeling (HTR) in a meta-RL training loop. HTR is agnostic to the underlying (off-policy) meta-RL algorithm; the agent architecture and/or training specifics (e.g., the encoder $\phi$, actor $\pi$ and $Q$-function neural networks shown in blue) can be modified independently of the relabeling scheme. HTR can also be performed in an 'eager' fashion at the data collection stage (as opposed to 'lazy' relabeling in the data sampling stage), see Section 3 for details.

Relabeling transitions under hindsight tasks serves as an additional source of supervision in learning the latent task embedding $\mathbf{z} \Leftarrow f(\mathbf{c})$ by enabling gradient-based optimization of $f$ in the absence of meaningful reward signal (in PEARL, this embedding is parameterized by the context encoder $q_\phi$). Similar to how HER generates an implicit curriculum of goals, HTR generates an implicit curriculum of tasks that gradually shifts from easier hindsight tasks towards the true training distribution $\mathcal{T}_{train} \sim p(\mathcal{T})$: initially, the agent is unable to recover reward signal on the true training tasks, but it can bootstrap the learning of an adaptation strategy by learning on easier hindsight tasks.

We outline our method for meta-training with Hindsight Task Relabeling in Algorithm 1 (task relabeling is only used during meta-training; meta-testing is unchanged). We evaluate our approach using PEARL as the off-policy meta-RL algorithm $A$, though the approach we describe is general to any off-policy meta-RL algorithm, and could potentially be integrated with on-policy context-based meta-RL algorithms (such as RL$^2$) via hindsight policy gradients (Rauber et al., 2019).

## 4.1 Algorithm Design

Two important considerations in adapting hindsight relabeling to meta-RL are (i) how to choose the hindsight task, and (ii) how to sample the transitions to be relabelled. Together these choices form the task relabeling strategy $S$ in Algorithm 1. In HER, transitions are relabelled with goals that are optimal at the trajectory-level (see Figure 1). The key difference between our method and HER is that our meta-RL relabeling scheme needs to construct batches of data that share the same pseudo-task (or real task), which is fundamentally different from standard HER. HER does not consider the setting where batches of trajectories are collected in different MDPs; in HER, transitions are relabeled independently with locally-optimal pseudo-goals, and batches of training data for the agent contain many different pseudo-goals (and real goals). In the meta-RL setting, the agent must estimate the task from a recent context of transitions with the same underlying task (unknown to the agent), and therefore relabeling should take into consideration more than a single trajectory; the agent explores an environment which has a fixed task $\mathcal{T} \sim p(\mathcal{T})$, and as it accumulates context it uses the context to identify the singular task and update its policy accordingly.

A naive application of hindsight relabeling to meta-RL would assign each transition in the batch its own hindsight task (e.g., give each $\tau$ in Figure 1b its own $\mathcal{T}'$). This is a simple and straightforward way to apply HER to meta-RL that may seem correct at the surface-level, however, this is bound to fail if the meta-RL training process is left unchanged, since the context encoder is trained to estimate the task from a collection of transitions generated in that task (not a batch of transitions generated

**Algorithm 1:** Hindsight Task Relabelling for Off-Policy Meta-Reinforcement Learning

---
**Require:** Off-policy meta-RL algorithm $A$, training tasks $\mathcal{T}_{1:T} \sim p(\mathcal{T})$, context encoder $\phi$, actor $\pi$, critic $Q$, relabelling function $R$, task relabelling strategy $S$, and task relabelling probability $K$

1   Initialize a replay buffer $B_i$ for each training task $\mathcal{T}_i$
2   **while** *meta-training* **do**
3      **for** *each $\mathcal{T}_i$* **do**
4         Collect data on $\mathcal{T}_i$ according to $A$ (e.g., by rolling out $\pi$ conditioned on $z \sim \phi$)
5         Add data to task replay buffer $B_i$
6      **for** *each $\mathcal{T}_i$* **do**
7         $p_h \sim \mathcal{U}(0,1)$
8         **if** $p_h \geq K$ **then**
9            Select a minibatch of transitions $b_i$ and hindsight task $\mathcal{T}_h$ using strategy $S$
10              (e.g., sample transitions from trajectory $t$, and select a state reached in $t$ for $\mathcal{T}_h$)
11            Relabel transitions in $b_i$ according to $\mathcal{T}_h$, i.e., $R(b_i, \mathcal{T}_h)$
12         **else**
13            Sample minibatch of transitions $b_i \sim B_i$
14         Compute gradients and update $\phi$, $\pi$, and $Q$ using $b_i$ according to $A$

---

across several distinct tasks). A similarly flawed approach is to generate a single hindsight task from a random transition in the batch, and then to relabel the entire batch of transitions with the same hindsight task (e.g., choose a $\mathcal{T}'$ Figure 1b using a random $\tau$ to relabel all $\tau$s with). The issue with this approach is that the hindsight task is only guaranteed to be optimal for a single trajectory, and may in fact be highly sub-optimal for other trajectories in the batch. We present two relabeling strategies, *Single Episode Relabeling (SER)* and *Episode Clustering (EC)*, that are inspired by the relabeling strategy used in Andrychowicz et al. (2017) but are specifically designed to work in the meta-RL setting.

## 4.2   Single Episode Relabeling (SER) strategy

In SER, a single episode (i.e., trajectory) is randomly selected to generate the hindsight task and sample transitions from (i.e., sample $N$ transitions from 1 trajectory, each relabelled under 1 new task). Despite the fact that this reduces the pool of context from the entire task buffer to a single episode, this setting closely resembles meta-testing (where context is drawn purely online), and we found this resampling strategy to work well in practice since it results in more relabelled transitions with non-zero reward. HTR using SER is outlined in Algorithm 1.

## 4.3   Episode Clustering (EC) strategy

An alternative strategy to Single Episode Resampling is to cluster trajectories that satisfied similar hindsight tasks into the same task buffers $B_h$, essentially creating additional task buffers for hindsight tasks that can be used for training in-place of the real task buffers $B_i$ (which are also sampled with some probability $K$). For example, in goal-reaching tasks, trajectories can easily be mapped to hindsight buffers with trajectories from separate trials by discretizing the state space and creating a buffer for each partition. In more complex tasks, alternative clustering approaches (e.g., learning-based) can be used to group together trajectories that can be relabeled with high reward conditioned on the same tasks. Episode clustering relabels a similar number of transitions with non-zero reward to the resampling strategy, but with less duplicate transitions per-batch. In practice, we found the SER strategy to be far less complex and similarly effective to the EC approach. See Section 5 for an empirical comparison between SER and EC, and the supplement for a more detailed comparison.

## 4.4   Comparison of HTR and HER

Both SER and EC are similar to the relabeling strategy used in HER in some respects, but differ in others: HER samples $N$ transitions from the full replay buffer, and relabels each transition independently ($N$ different goals). HTR with SER samples 1 trajectory, and samples $N$ transitions from that trajectory all relabelled with 1 new task. HTR with EC samples $N$ transitions from 1

hindsight replay buffer, where each transition in that buffer has been relabeled with 1 new task. In HER, the hindsight goal for a given transition is chosen by selecting a random transition that occurs after the sampled transition in the same trajectory. In HTR, hindsight tasks are assigned in the same way as HER, but a single hindsight task is applied to an entire batch of transitions.

An important distinction between HER and HTR is that in HER, bootstrapping training with hindsight goals does not change the optimal policy, since a goal-conditioned policy with sufficient capacity should (in theory) be able to represent an optimal policy for every goal (including both original goals and hindsight goals). In contrast, the optimal exploration strategy for a particular meta-RL environment depends on its true task distribution, and bootstrapping training with hindsight tasks may expand the training task distribution in a manner that changes the optimal exploration strategy. One way to benefit from HTR's bootstrapping while ensuring the final learned strategy is optimal for the original task distribution is to only relabel data from tasks on which the agent has never received (non-zero) reward. Similarly, another solution is to anneal the relabelling probability $K$ to zero during meta-training. Neither method was required to achieve the results shown in this paper, however, we expect they may be useful if applying HTR to other meta-RL environments.

### 4.5 Limitations

The key limitation of our method is that it assumes trajectories can be relabeled under new task (i.e., a mapping $\mathbf{s} \to \mathcal{T}$ or $\tau \to \mathcal{T}$ exists), which is a reasonable assumption for goal-reaching environments (as studied in Rakelly et al. (2019); Gupta et al. (2018b); Andrychowicz et al. (2017)), but may be significantly more challenging in other environments. Imagine a complex robotic manipulation environment (e.g., retrieving an object from a drawer) where reward is sparse and only received upon successful completion of the entire task; in this scenario, relabeling the zero-reward transitions generated by an initial random policy into useful signal for training an optimal policy is less straightforward than the goal-reaching setting, and may require explicit sub-task specification using domain expertise.

Similar to how HER is not necessarily only for goal-reaching *goal-conditioned* RL, HTR is not necessarily only for goal-reaching *meta*-RL, and if a good relabeling function exists then either approach can be successfully applied in their respective setting (HER for goal-conditioned RL, HTR for meta-RL). However, as in the original HER work, we focus our experiments on goal-reaching tasks, where the reward function is a simple function of the state and can be easily repurposed for relabeling. It may be possible to relax this assumption and extend HTR to more families of tasks by employing more complex task relabeling schemes, e.g., by extending work on inverse RL for hindsight relabeling (Li et al., 2020; Eysenbach et al., 2020) to the meta-RL setting, or by carefully engineering reward functions specific to each environment, however we leave this to future work.

Additionally, our method adds a new hyperparameter $K$ to the underlying off-policy meta-RL algorithm, which may need to be tuned for optimal results. The specification and tuning of a relabeling probability $K$ is also required in standard HER (see Andrychowicz et al. (2017)).

## 5   Experiments

In evaluating our proposed method, we aim to answer the following questions. (i) Can Hindsight Task Relabeling enable (meta-)learning of adaptation strategies in challenging sparse reward environments, where existing meta-RL methods fail without shaped reward? (ii) How do the adaptation strategies learned using Hindsight Task Relabeling compare to those learned using shaped reward functions? (iii) How do key implementation choices (such as relabeling probability $K$) affect its performance?

### 5.1   Environments

We evaluate our method on a suite of sparse reward environments based those proposed by Gupta et al. (2018b) and Rakelly et al. (2019) (see Figure 2). In prior work, each environment exposes two reward functions: a dense reward function used during meta-training, and a sparse reward function used during meta-testing. The key difference in our experimental setup is that we consider the setting where sparse reward is used both during meta-testing *and meta-training*. In each environment, a set of 100 tasks is sampled for meta-training, and a set of 100 tasks is sampled from the same task distribution for meta-testing. The environments were each modified to increase their difficulty, such

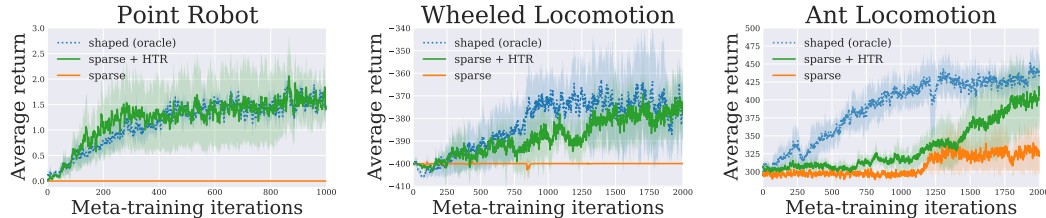

Figure 4: Evaluating adaptation to train tasks progressively *during* meta-training. Y-axis measures average sparse return during adaptation throughout meta-training (shaded std dev), though the oracle is still trained using dense reward. Conventional meta-RL methods struggle to learn using sparse reward. Hindsight Task Relabeling (HTR) is comparable to dense reward meta-training performance.

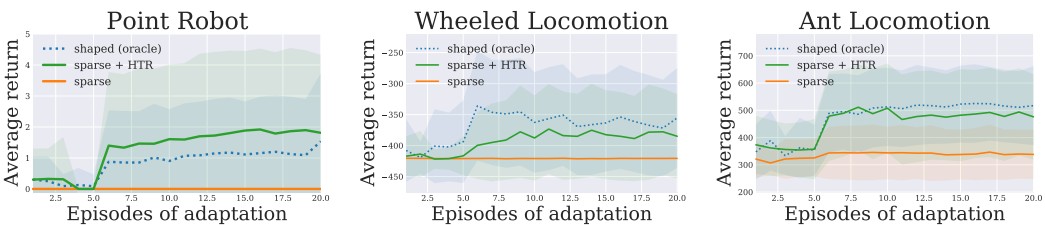

Figure 5: Evaluating adaptation to test tasks *after* meta-training. Y-axis measures average (sparse) return during adaptation using context collected online, using sparse reward only. Adaptation strategies learned with Hindsight Task Relabeling (HTR) generalize to held-out tasks as well as the oracle which is learned using shaped reward functions. Without HTR or access to a shaped reward during meta-training, the agent is unable to learn a reasonable strategy.

that random exploration is far less likely to encounter sparse reward. Refer to the supplement for further details on the experimental setup.

- *Point Robot*. A point robot must navigate a 2D plane to different goals located along the perimeter of a half-semicircle. The state includes the robot's coordinates but does not include the goal, therefore the agent must explore the environment to discover the goal. In the dense reward variant of the environment, the reward is the negative $L2$ distance to the goal. In the sparse reward variant, reward is given only when the agent is within a short distance to the goal. An example optimal exploration strategy in the sparse reward variant is to efficiently traverse the perimeter of the semicircle until the goal is found.

- *Wheeled Locomotion*. To test if our approach generalizes to more complex state and action spaces, we use several continuous control environments. In the wheeled locomotion environment, the agent must navigate to the goal distribution by controlling two wheels independent to turn. Similar to the point robot, the dense reward function is the negative distance to the goal, while the sparse reward function only provides reward signal when the agent is nearby the goal.

- *Ant (Quadruped) Locomotion*. This environment requires controlling a quadruped ant with a high-dimensional state and action space. The task distribution and reward functions are the same as the point robot and wheeled locomotion environment, however, exploration requires coordinating movement with four legs to navigate to specific locations.

## 5.2   HTR enables meta-training using only sparse reward

To answer question (i), we compare our hindsight task relabeling approach (using PEARL as the base meta-RL algorithm) to standard PEARL, as well as an oracle PEARL which uses the dense reward function during training. Our oracle baseline is equivalent to the standard meta-RL setup in the 'sparse reward' experiments in Gupta et al. (2018b) and Rakelly et al. (2019), despite it never training on the sparse reward function. Note that the term 'oracle' is somewhat of a misnomer, since training on a proxy dense reward function designed to aid the learning of an RL agent (to be evaluated

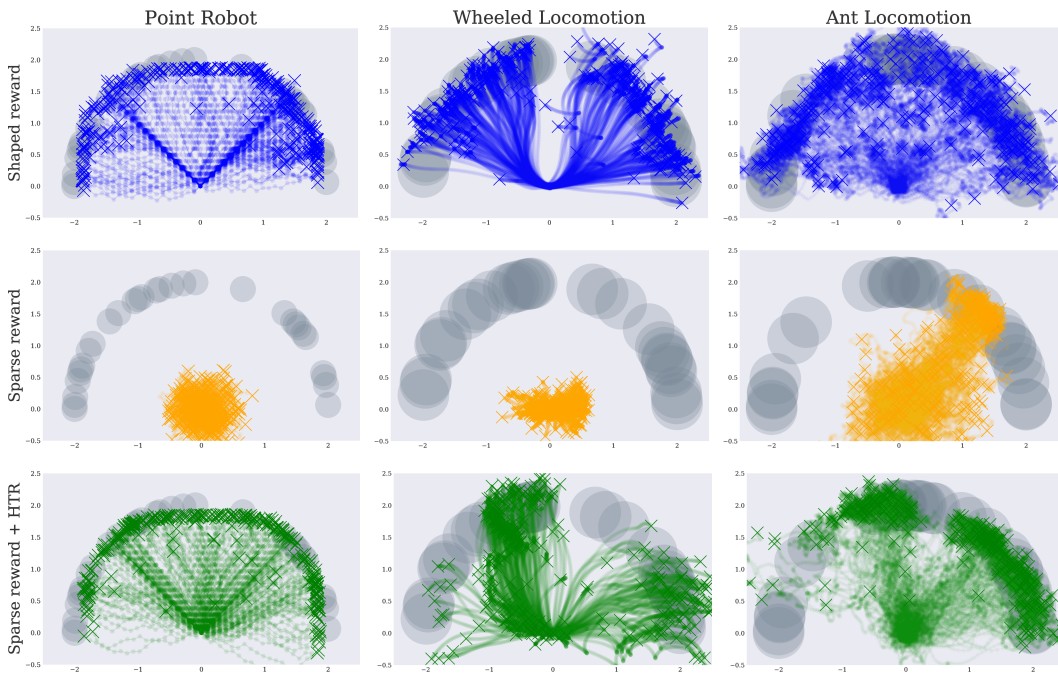

Figure 6: Visualizing exploration behavior learned during meta-training using 300 pre-adaptation trajectories (i.e., sampled from the latent task prior). In the sparse reward setting, without HTR (middle row) the agent is unable to learn a meaningful exploration strategy and appears to explore randomly near the origin. With HTR (bottom row), the agent learns to explore near the true task distribution (grey circles), similar to an agent trained with a shaped dense reward function (top row).

on the sparse reward function) is inherently optimizing a different objective. Optimizing an RL agent for a proxy objective has been shown to lead to inadvertently bad performance on the true objective (Clark & Amodei, 2016), and therefore it is often better to directly optimize for the true objective (e.g., the original sparse reward function) when possible.

As discussed in prior work, meta-learning on sparse reward environments proves extremely challenging: in Figures 4 and 6, we see that PEARL is unable to make learn a reasonable policy just using sparse rewards in the same amount of time it takes the oracle (PEARL trained on dense reward) and HTR (PEARL trained on sparse reward using Hindsight Task Relabelling) to converge. In all of our tested environments, HTR is not only comparable to using shaped reward during meta-training, but it also can learn adaptation strategies that perform as well as the oracle during meta-testing on the sparse reward environment. In Figure 5, we see that the adaptation strategies learned with Hindsight Task Relabeling are similarly effective to those learned by the oracle, despite the oracle having access to a dense reward function during meta-training.

Given an indefinite amount of meta-training time, or an easier environment configuration (e.g., a shorter goal distance), PEARL should be able to learn a similarly optimal strategy to the oracle and HTR only using sparse reward. If PEARL *is* able to successfully learn only using sparse rewards, HTR can be used to improve sample efficiency during meta-training. In the case that the amount of time or computational resources needed to train purely on sparse rewards is intractable, HTR is extremely effective at learning adaptation strategies comparable to using a hand-designed dense reward function.

Additionally, as mentioned in Section 2, in many sparse reward settings designing a dense reward function to aid training is infeasible or undesirable, in which case HTR is a strong alternative to reward engineering. Because HTR does not train on a proxy reward, there is no opportunity for a mismatch between reward functions at meta-training and meta-testing: HTR optimizes for the true reward function on a superset of tasks $\mathcal{T}_{hindsight} \cup \mathcal{T}_{train} \sim p(\mathcal{T})$, whereas using a shaped reward optimizes for a proxy reward function on the true set of training tasks $\mathcal{T}_{train} \sim p(\mathcal{T})$.

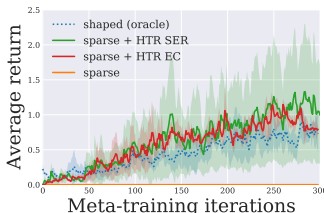
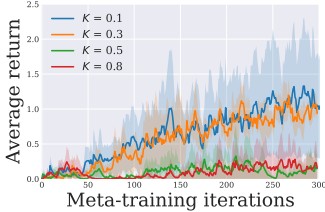
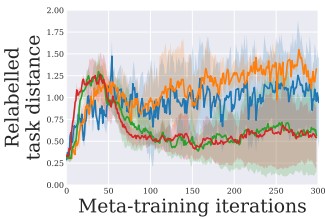

Figure 7: Comparing HTR with SER vs EC on Point Robot.

Figure 8: Average return when varying $K$ on Point Robot.

Figure 9: Average task distance when varying $K$ on Point Robot.

## 5.3 Varying key hyperparameters

In our experiments we found a relatively low relabeling probability (e.g., $K = 0.1$ and $0.3$) was often most effective for HTR (see Figure 8). $K = 0.1$ means that relabelled data from a hindsight task is roughly as likely to be used for training as data from any of the 100 tasks in $\mathcal{T}_{train}$: $K = 0.1$ corresponds to using far less relabelled data for training than in conventional HER: in contrast, Andrychowicz et al. (2017) relabel over 80% of sampled transitions. We hypothesize that a low $K$ works best for HTR since it provides enough reward signal to bootstrap learning, without biasing the task distribution too far from the ground truth distribution; in Figure 9, we can see that both $K = 0.5$ and $0.8$ converge on a mean relabelled task distance of $0.5$, whereas $K = 0.1$ and $0.3$ are closer to the ground truth task distance of $2.0$.

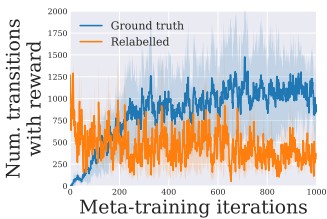

Figure 10: Relative reward signal from hindsight vs ground truth tasks using Point Robot.

In both HER and HTR, the relabeling probability is an important hyperparameter that should be tuned for optimal performance. In the original HER paper, Andrychowicz et al. (2017) reported that a relabelling probability of 80% or 90% worked best on their three goal-conditioned RL environments. For both algorithms, there exists a range of relabelling probabilities (alternatively a range of ratios of original-to-relabelled data) where the relabeling algorithm works well, and a range of values where the algorithm performs poorly (the HER paper did not include results for under 50% relabelled data, which may include more negative results).

In Figure 9, we see that the average distance reached by the policy during meta-training gradually increases over time, increasing the average relabelled task distance. Hindsight tasks used for relabelling initially correspond to easily achievable tasks, and shift towards the ground truth task distribution. Initially, the only reward signal available for training comes from the relabelled transitions, however, as training progresses the agent is able to recover true reward signal from the environment (see Figure 10) and the hindsight tasks are more likely to resemble the ground truth tasks.

As described in Section 4, we found that EC generally performed similarly to SER (see Figure 7), despite being significantly more complex to implement and requiring additional tuned hyperparameters. We expect that a more significant performance difference between the two approaches may become apparent on different sparse reward meta-RL tasks, or through experimenting with more general episode clustering approaches than the state space discretization used in this work.

## 6 Conclusion

In this paper, we propose a novel approach for meta-reinforcement learning in sparse reward environments that can be incorporated into any off-policy meta-RL algorithm. The sparse reward meta-RL setting is extremely challenging, and existing meta-RL algorithms that learn adaptation strategies for sparse reward environments often require dense or shaped reward functions during meta-training.

Our approach, Hindsight Task Relabeling (HTR), enables learning adaptation strategies in challenging sparse reward environments without engineering proxy reward functions. HTR relabels data collected on ground truth meta-training tasks as data for achievable hindsight tasks, which enables meta-training on the original sparse reward objective. Not only does HTR allow for learning adaptation strategies in challenging sparse reward environments without reward engineering, but it also generates adaptation strategies comparable to prior approaches that use shaped reward functions.

## Acknowledgements

We thank Katelyn Gao for providing valuable feedback and insights on our work. This project is in part based upon work supported by the National Science Foundation under NSF CISE Expeditions Award CCF-1730628, Berkeley Deep Drive, and gifts from Amazon Web Services, Ant Group, Ericsson, Facebook, Futurewei, Google, Intel, Microsoft, Scotiabank, and VMware. Any opinions, findings, and conclusions or recommendations expressed in this paper are those of the authors and do not necessarily reflect the views of the National Science Foundation, Berkeley Deep Drive, and other sponsors.

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
