# A  Experimental Setup

## A.1  Computing Infrastructure

Our experiments were run on NVIDIA Titan Xp GPUs and Intel Xeon Gold 6248 CPUs on a local compute cluster. When running on a single GPU, the time required to run a single experiment ranges from approximately a day (e.g., Point Robot) to several days (e.g., MuJoCo locomotion tasks).

## A.2  Hyperparameters

In our experiments we perform a sweep over key hyperparameters including task relabelling probability $K$ (we swept over values of [.1, .3, .5, .8, 1.]). Our reported results use the best performing $K$, with all other PEARL hyperparameters set to the same as in the PEARL baselines. Results are averaged across five random seeds.

For PEARL, we use a latent context vector $\mathbf{z}$ of size 5, a meta-batch size of 16 (number of training tasks sampled per training step). The context network is has 3 layers with 200 units at each layer. All other neural networks (the policy network, value and Q networks) have 3 layers with 300 units each. The learning rate for all networks is $3\mathrm{e}^{-4}$.

## A.3  Reward Functions

In the *Point Robot* environment, the dense reward is the negative distance to the goal, and the sparse reward is a thresholded negative distance to the goal, rescaled to be positive: $-dist(robot, goal) + 1$ if $dist(robot, goal) < 0.2$, 0 otherwise. In the two locomotion environments from Gupta et al. (2018b) (*Wheeled Locomotion* and *Ant Locomotion*), the reward function includes a dense goal reward (negative distance to the goal), as well as a control cost, contact cost, and survive bonus. In our variation of these environments, we modify the goal reward to be sparse (for both meta-training and meta-testing), but leave the remaining reward terms unmodified.

## A.4  Changing the Distance to Goal

In our experiments, we used a goal distance set far enough from the origin such that random exploration is unlikely to lead to sparse reward (therefore requiring either a dense reward function or Hindsight Task Relabelling to make progress during meta-training). If the distance to the goal is reduced to a point where sparse reward is easily found through random exploration, meta-training is possible on the sparse reward function without needing Hindsight Task Relabelling (see Figure 11).

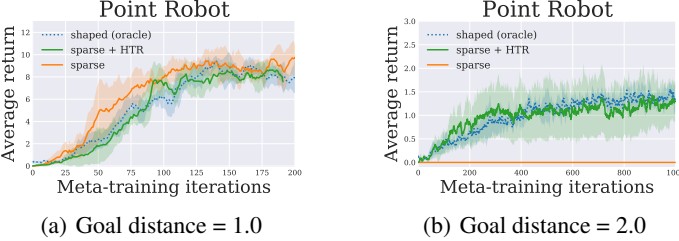

(a) Goal distance = 1.0    (b) Goal distance = 2.0

Figure 11: Meta-training on *Point Robot* with varying goal distances. If the distance to the goal is short enough for random exploration to lead to sparse reward, meta-training is possible using only the sparse reward function. Once this is no longer the case, meta-training is only possible with a proxy dense reward function, or by using Hindsight Task Relabelling on the original sparse reward function.

# B  Algorithm Specifics

## B.1  Sample-Time vs Data Generation Relabelling

There are two stages in the off-policy meta-RL algorithm meta-training loop in which data relabelling can occur: during data collection (when the policy collects data by acting in the environment with

a set task), and during training (when samples from the replay buffer are used to update the policy, value function, encoder, etc.). We refer to the former as 'data generation relabelling' (or 'eager' relabelling, since the new labels are computed before they are needed), and 'sample-time relabelling' (or 'lazy' relabelling, since the new labels are computed on-the-fly when the are needed to compute gradients). The Single Episode Relabelling (SER) strategy described in the main text is a sample-time relabelling approach, whereas the Episode Clustering (EC) strategy is a data generation relabelling approach. See Figure 12 and Algorithm 2 for an outline of HTR with data generation relabelling (differences to HTR with sample-time relabelling are highlighted in blue).

## B.2 Single Episode Relabelling Implementation Details

Single Episode relabelling (SER) samples a single episode from the task replay buffer, samples $N$ transitions from that episode and rewrites the rewards for each sampled transition under a new hindsight task. These transitions are used for both the context batch and the RL batch in PEARL. We found that sampling transitions from the same episode for both the context batch and RL batch is important for performance - the alternative (sampling an episode $e$ from task buffer $b$, choosing a hindsight task $t$, sampling the context batch from $e$ relabelled under $t$, then sampling an RL batch from the entire buffer $b$ relabelled under $t$) often results in low reward in the RL batch, despite having high reward in the context batch. This is because the replay buffer (particularly at the beginning of training) contains a diverse set of trajectories each with a different optimal hindsight task; sampling from the entire task buffer but relabeling under single hindsight task optimal only for a specific trajectory in the buffer can easily lead to a batch of transitions with all zero reward.

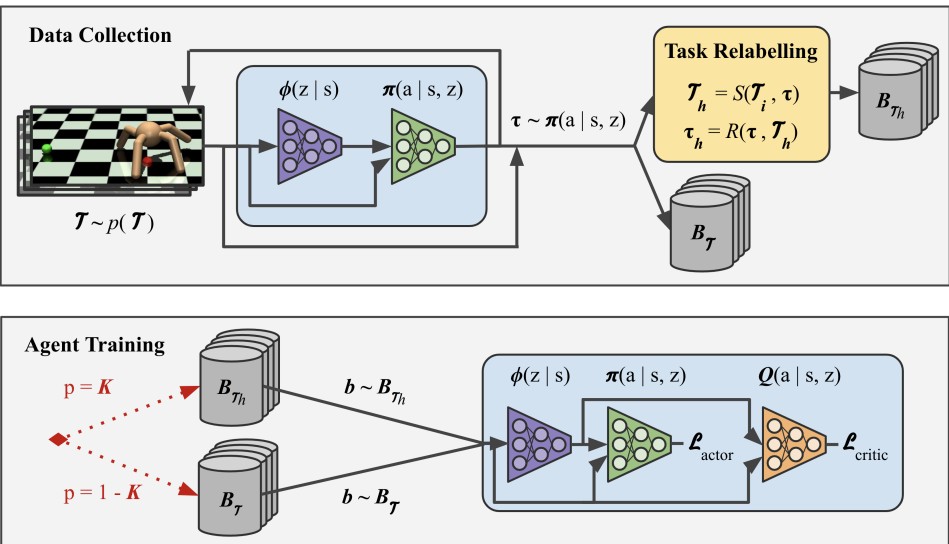

Figure 12: Illustration of Hindsight Task Relabeling (HTR) using Episode Clustering (EC) in a meta-RL training loop, where relabelling occurs at the data collection stage.

## B.3 Episode Clustering Implementation Details

Episode Clustering requires maintaining a mapping from trajectories to tasks in the real task buffers $B$, e.g., by tracking the episodes across multiple task buffers, or by storing the relabelled transitions in an additional set of hindsight buffers $B_h$ (which requires additional memory). In meta-RL environments that consider goal-reaching tasks, simple discretization is a reasonable assumption (e.g., via a grid) that allows for an easy way to map each trajectory to a hindsight tasks, however, such discretization requires additional hyperparameters and tuning. More importantly, Episode Clustering requires additional tracking to ensure that the distribution of transitions in the hindsight buffers (implicit or explicit) remains similar to the real task buffers, which contain a significant amount of low-reward exploratory trajectories.

In practice, we found the SER strategy to be far less complex and similarly effective to the EC approach, however the benefits of EC may be more apparent in non-goal-reaching tasks. SER has the

significant benefit of requiring far less hyperparameter tuning - for SER, only relabelling probability needs to be tuned (similar to HER), whereas EC requires either implementing a method for clustering (e.g., discretization) and tuning all the relevant hyperparameters for the clustering approach (there may be significantly more hyperparameters beyond $K$). We found the balancing of the exploration vs exploitation trajectories in the hindsight replay buffers to be important in getting EC to work well in practice. Meanwhile, SER is relatively simple to implement on top of the existing PEARL (meta-training) sampling routine.

EC samples context and RL batches the same way as in standard PEARL: a batch of recent transitions from the buffer is sampled for context, and a batch of transitions from the entire buffer is sampled for the RL batch. The key difference between HTR with EC and PEARL (at the sampling stage) is that a real task buffer is swapped out with a hindsight task buffer according to probability $K$. The other key difference compared to PEARL is at the data collection stage, where relabelled transitions are added to the hindsight task buffers.

---

**Algorithm 2:** HTR with data generation relabelling (i.e., Episode Clustering)

---

   **Require:** Off-policy meta-RL algorithm $A$, training tasks $\mathcal{T}_{1:T} \sim p(\mathcal{T})$, context encoder $\phi$,
             actor $\pi$, critic $Q$, relabelling function $R$, task relabelling strategy $S$, and task
             relabelling probability $K$

1  Initialize a replay buffer $B_i$ for each training task $\mathcal{T}_i$
2  **while** *meta-training* **do**
3     **for** *each $\mathcal{T}_i$* **do**
4         Collect data $d$ on $\mathcal{T}_i$ according to $A$ (e.g., by rolling out $\pi$ conditioned on $z \sim \phi$)
5         Add data $d$ to task replay buffer $B_i$
6         **for** *each trajectory $t \in d$* **do**
7             Select hindsight task $\mathcal{T}_h$ using strategy $S$ (e.g., select a state reached in $t$)
8             Relabel transitions in $t$ according to $\mathcal{T}_h$, i.e., $R(t, \mathcal{T}_h)$
9             Add the modified $t$ to hindsight task buffer $B_{\mathcal{T}_h}$ by mapping the $\mathcal{T}_h$ to a discrete
               buffer index (e.g., by discretizing the state space)
10    **for** *each $\mathcal{T}_i$* **do**
11       $p_h \sim \mathcal{U}(0,1)$
12       **if** $p_h \geq K$ **then**
13          Select a random hindsight task buffer $B_{\mathcal{T}_h}$
14          Sample minibatch of transitions $b_i \sim B_{\mathcal{T}_h}$
15       **else**
16          Sample minibatch of transitions $b_i \sim B_i$
17       Compute gradients and update $\phi$, $\pi$, and $Q$ using $b_i$ according to $A$

---

### B.4   Time and Space Complexity

HTR increases the time complexity of the original PEARL algorithm by a constant which is determined by the time complexity of the relabelling routine. In our reference implementation of HTR, the relabelling routine has limited overhead (it is similar to calling the environment's own reward function, which itself is called at every timestep of the environment); in practice, we found the HTR version of PEARL to have similar runtime as the original PEARL. Note that this is the same overhead that the original HER algorithm adds on top of its underlying goal-conditioned RL algorithm (e.g., goal-conditioned DDPG). The added overhead from HTR increases linearly with hyperparameter $K$.

Since the Single Episode Relabelling strategy relabels at sample-time (similar to HER), it has no added space complexity (i.e., it is equivalent to the space complexity of PEARL). The Episode Clustering strategy relabels during data generation, and can be implemented using pointers (minimal space overhead) or with additional replay buffers for each cluster. Our reference implementation uses additional replay buffers, and therefore space complexity grows linearly with the number of buffers.

### B.5   Theoretical Analysis of HTR

The HER paper (Andrychowicz et al., 2017) does not include formal guarantees on performance, yet the method of great practical interest due to its strong empirical performance. Similarly, we do

not provide any formal guarantees on performance for HTR, however we demonstrate that HTR enables learning on several sparse reward meta-RL environments that previously required dense reward. Deriving formal guarantees for HTR would likely require first deriving formal guarantees for HER, then generalizing those results to the meta-RL setting. We consider this outside the scope of this paper and a potential focus of future work.

## C   Broader impacts

Deep reinforcement learning has been successfully applied to numerous application areas: robotics (e.g., human-robot interaction, self-driving cars), infrastructure (e.g., resource management, traffic light control), life sciences, advertising, finance. Each of these application areas has a broad range of societal impacts and ethical considerations. Concerns that arise with deploying deep learning models (such as brittleness to adversarial attacks, data privacy, lack of interpretability/explainability, etc.) also affect deep RL and (deep) meta-RL. Our work aims to improve state-of-the-art meta-reinforcement learning algorithms, and we believe our work will enable meta-RL to be applied to and solve new problems. We encourage researchers and practitioners looking to apply our research to consider the limitations of deep RL and meta-RL methods, and thus the implications of learning agents or policies using such methods.