# OpenReview forum: "Hindsight Task Relabelling: Experience Replay for Sparse Reward Meta-RL"
_NeurIPS.cc/2021/Conference — NeurIPS 2021 Poster_

### Official Review · Reviewer_iJzp · 2021-07-15

**Rating:** 6
**Confidence:** 3

**Summary:**

This work extends HER to meta-RL setting where the task parameters are not provided to the agent. The authors propose two different ways to relabel transitions and demonstrate their effectiveness through experiments. On goal reaching tasks, the performance of HTR on sparse reward setting is comaparable to baseline under dense reward setting.

**Limitations And Societal Impact:**

The authors have addresed the limitations adequately.

**Main Review:**

Originality: Low. HTR is a direct extension of HER. The authors make minor modifications to HER to adapt to meta-RL setting.

Quality: Good. While empirical and straightforward, the proposed methods (SER and EC) are sounding and achieved good performance on widely-used environments. However, it would be more convincing if the authors could provide a more formal description of HTR.

Clarity: Medium. The story is well-written but some details of the algorithm is unclear. E.g. in l.199, how to determine the new hindsight task is not answered until l.221.

Significance: Medium. While sparse reward+HTR achieved near-dense reward setting performance in experiments, the algorithm is limited to goal reaching task and it remains questionable whether it will work for for complex meta-RL tasks.

Major concerns:
 1. The authors claim that "HTR generates an implicit curriculum of tasks that gradually shifts from easier hindsight tasks towards the true training distribution". I believe this is overclaimed and not supported by the experiments in the supplementary. The distribution of the generated hindsight tasks simply follows the current policy.
2. Compared with SER which is a modified version of HER, EC is a more sounding approach to me. I believe that the potential of EC is underclaimed in the paper. If I understood correctly, the EC method allows mixture of transitions coming from different episodes and could therefore train the context encoder with much more samples. In simple task such as goal reaching, it could be hard to demonstrate the effectiveness of EC since the context encoder is simply an exclusion of visited coordinates. I suggest that the authors add more discussion on this or even experiments, if possible. If the authors could use SER as a baseline and demonstrate how EC works better than SER in more complex settings, this paper would definitely be a valuable one.

Minor issues:
 1. Algorithms are too shortened that it is hard to tell the difference between Alg. 1 in the main text and Alg. 1 in the supplementary. According to l.216~219, in SER all transitions in a batch comes from the *same* trajectory and share the same hindsight task. However, this is not outlined in the algorithm.
 2. The authors claimed several times that HTR creates a "dense" reward. This may not be true since the agent is still rewarded at goal states only.
 3. Introduction section missing.

Questions:
 1. In supp. Figure 2, why is the num. of rewarded relabelled transitions decreasing?

**Time Spent Reviewing:**

5 hours

---

> ### Author Response · Authors · 2021-08-10
> **Response to Reviewer iJzp**
>
> Thank you for your valuable feedback.
>
> > *The authors make minor modifications to HER to adapt to meta-RL setting*
>
> To clarify any potential misunderstanding - HTR has significant and fundamental differences compared to HER, which are summarized in lines 214-221. HER is a method for goal-conditioned RL; the agent receives the goal as input, and there is no adaptation stage where the goal or task must be inferred from context. This allows the relabelling to happen at the transition-level, i.e., every transition sampled for a batch of training is independently relabelled with its own goal. HTR is a method for meta-RL, and therefore must relabel experiences in a way that enables the agent to learn to infer a goal or task from context, and exploit that knowledge to receive reward. This constraint means the relabelling needs to happen at the episode-level (or trial-level, where a trial consists of multiple episodes on the same task), which requires significant modifications to the relabelling algorithm and implementation. This fundamental difference in relabelling schemes is crucial in applying the relabelling ideas from HER relabelling effectively in the meta-RL setting; as we discussed in Section 3.1 (lines 183-196), applying HER-style transition-level relabelling to a meta-RL algorithm such as PEARL (which might constitute a “minor modification” of HER to the meta-RL setting) may seem reasonable at first glance, but does not yield good results in practice.
>
> > *The authors claim that "HTR generates an implicit curriculum of tasks that gradually shifts from easier hindsight tasks towards the true training distribution". I believe this is overclaimed and not supported by the experiments in the supplementary. The distribution of the generated hindsight tasks simply follows the current policy.*
>
> We agree with this critique and will revise Appendix A.1 to state that the average distance reached by the policy during meta-training gradually increases over time, therefore increasing the average relabelled task distance gradually.
>
> > *Compared with SER which is a modified version of HER, EC is a more sounding approach to me. I believe that the potential of EC is underclaimed in the paper. If I understood correctly, the EC method allows mixture of transitions coming from different episodes and could therefore train the context encoder with much more samples. In simple task such as goal reaching, it could be hard to demonstrate the effectiveness of EC since the context encoder is simply an exclusion of visited coordinates. I suggest that the authors add more discussion on this or even experiments, if possible. If the authors could use SER as a baseline and demonstrate how EC works better than SER in more complex settings, this paper would definitely be a valuable one.*
>
> Your understanding is correct - EC allows mixtures of transitions from different trajectories (that all satisfied a similar hindsight task) to be grouped together into a single buffer and later sampled during meta-training.
>
> We will revise Section 3.1 to expand on the potential benefits of EC described on lines 210-211, as well as mention that the benefits of a method such as EC may become more apparent in non-goal-reaching tasks. We will also revise Figures 4 and 5 to include SER and EC side-by-side to help frame the discussion in Section 3.1 on SER vs EC.
>
> SER has the significant benefit of requiring far less hyperparameter tuning - for SER, only relabelling probability $K$ needs to be tuned (similar to HER), whereas EC requires either implementing a method for clustering (e.g., discretization) and tuning all the relevant hyperparameters for the clustering approach (there may be significantly more hyperparameters beyond $K$). There are also more important implementation details for the EC method as described in Appendix C.2. Specifically, we found the balancing of the exploration vs exploitation trajectories in the hindsight replay buffers (lines 50-53) to be very important in getting EC to work well. Meanwhile, SER is relatively simple to implement on top of the existing PEARL (meta-training) sampling routine.
>
> > *Algorithms are too shortened that it is hard to tell the difference between Alg. 1 in the main text and Alg. 1 in the supplementary. According to l.216~219, in SER all transitions in a batch comes from the same trajectory and share the same hindsight task. However, this is not outlined in the algorithm.*
>
> We agree that this should be clarified in the algorithm boxes, and will revise Algorithm 1 line 9 (in the main text) to include an additional line expanding on how transitions are sampled using the SER strategy. Similarly, we will expand Algorithm 1 line 7 (in the appendix) to describe how transitions are relabelled and assigned to different hindsight task buffers.
>
> > *The authors claimed several times that HTR creates a "dense" reward. This may not be true since the agent is still rewarded at goal states only.*
>
> In these contexts, we mean that HTR relabels a batch of transitions with all zero reward into a batch of transitions with a significant amount of non-zero reward transitions (creating a “dense” reward). However we agree that this usage of the word “dense” is confusing (given the discussion on sparse vs dense/shaped reward functions) and may be incorrect, since some of the relabelled transitions may not have reward even after relabelling. We will revise these sentences (e.g., line 202: “results in a dense reward signal”) to use different terminology and avoid potential confusion.
>
> > *In supp. Figure 2, why is the num. of rewarded relabelled transitions decreasing?*
>
> At the beginning of training, many trajectories start and end near the origin, therefore many selected hindsight tasks are also near the origin, and transitions sampled from the trajectories for HTR have a high probability of being relabelled with non-zero reward. As training progresses, trajectories and their associated hindsight tasks expand further from the origin, and therefore the hindsight tasks still provide non-zero reward but for a smaller portion of transitions within each trajectory (so the total number of relabelled transitions with non-zero reward decreases).

---

### Official Review · Reviewer_brT1 · 2021-07-16

**Rating:** 6
**Confidence:** 4

**Summary:**

This paper describes an extension of Hindsight Experience Replay (HER) to a meta-RL setting with task inference.
Like HER, trajectories are relabelled with rewards as if they had achieved a goal. Unlike HER, the goal cannot be explicitly relabelled. Instead, a context is provided from which the agent attempts to learn a task embedding: this task inference is also trained using the reward-relabelled data.

**Limitations And Societal Impact:**

I discuss one potentially important limitation in the main review.

**Main Review:**

This paper is a neat extension of the nice HER idea to a task-inference setting. The main idea makes sense, the paper is fairly well written, and the results show a substantial qualitative change in performance compared to a more standard use of a sparse reward. Since it’s a (relatively) incremental extension of HER to a new setting, I can’t rate the originality or significance of the work as extremely high, but I think it’s a nice paper overall.

I have one major concern about the method that deserves some further discussion, which is about the task distribution induced by the hindsight relabelling.
In HER, a biased goal distribution is not asymptotically problematic if the model has sufficient capacity, eventually it can represent the optimal solution for every goal (although certainly the relabelling bias will impact generalisation in practice).
In HTR however, the optimal meta-learned exploration strategy depends on the true task distribution, as does the optimal task inference strategy.
The authors seem to be somewhat aware of this issue from the brief comment in line 309 in discussion of the K hyperparameter, but it seems like a more fundamental qualitative difference between the HER and HTR settings that deserves further discussion, unless I’ve misunderstood something.

My other comments are more minor on exposition and experiments:
I found the paragraph starting line 183 quite confusing; the ‘naive’ alternatives felt a bit like strawmen, or maybe they rest on assumptions that are more implementation-level, e.g. about what is/isn’t being done in a single batch. For example, I don’t see a problem with relabelling each transition in a batch, as long as they also each get their own associated context (this might be inefficient or undesirable in practice?).
I’d maybe recommend presenting the two approaches that ‘make sense’ first, and then contrasting with alternatives, or trying to make a bit more clear why one might consider the 'naive' approaches first.

For the experiments; while I understand that the point of the method is to show a step change from the inability to learn with a sparse reward, it’s a little unsettling to see a baseline method that completely flatlines on all tasks. It would be encouraging to see a simple experiment with e.g. a point-mass that needn’t travel as far to achieve a sparse reward (with the expectation that HTR increases sample efficiency, but the baseline clearly works).


**Time Spent Reviewing:**

4

---

> ### Author Response · Authors · 2021-08-10
> **Response to Reviewer brT1**
>
> Thank you for your valuable feedback.
>
> > *In HER, a biased goal distribution is not asymptotically problematic if the model has sufficient capacity, eventually it can represent the optimal solution for every goal (although certainly the relabelling bias will impact generalisation in practice). In HTR however, the optimal meta-learned exploration strategy depends on the true task distribution, as does the optimal task inference strategy. The authors seem to be somewhat aware of this issue from the brief comment in line 309 in discussion of the K hyperparameter, but it seems like a more fundamental qualitative difference between the HER and HTR settings that deserves further discussion, unless I’ve misunderstood something.*
>
> We agree that this is an important discussion point, and we will revise Section 3.1 to include a separate subsection for an in-depth discussion on this difference between HER and HTR.
>
> As you described, the optimal exploration and task inference strategy will depend on the true task distribution, yet with HTR the task distribution used for training includes tasks from the true task distribution, as well as additional hindsight tasks that might lie outside the true task distribution. The motivation behind relabeling is to provide the training signal required to bootstrap training in the absence of non-zero reward (e.g., in a challenging sparse reward environment), however, once an agent begins to receive reward from the ground truth tasks on its own, the need for relabelled hindsight tasks is diminished.
>
> One way to benefit from HTR’s bootstrapping while ensuring the final learned strategy is optimal for the original task distribution is to only relabel data from tasks on which the agent has never received (non-zero) reward. Similarly, another solution is to anneal the relabelling probability K to zero during meta-training. In some situations, the expanded training task distribution can be beneficial for the agent: although the typical meta-RL problem setup assumes the training tasks and testing tasks are drawn from the same distribution, if this assumption is broken, then the expanded training task distribution induced through relabeling may help the agent generalize to the test tasks.
>
> > *I found the paragraph starting line 183 quite confusing; the ‘naive’ alternatives felt a bit like strawmen, or maybe they rest on assumptions that are more implementation-level, e.g. about what is/isn’t being done in a single batch. For example, I don’t see a problem with relabelling each transition in a batch, as long as they also each get their own associated context (this might be inefficient or undesirable in practice?). I’d maybe recommend presenting the two approaches that ‘make sense’ first, and then contrasting with alternatives, or trying to make a bit more clear why one might consider the 'naive' approaches first.*
>
> The initial proposition of “naive” alternatives is indeed based on implementation-level details - each proposed “naive” method in Section 3 is supposed to be a reasonable (but ultimately incorrect) way to extend HER to the meta-RL setting. We will revise Section 3.1 to make it clear that the “naive” alternatives are simple and straightforward ways to apply HER to meta-RL that may seem correct at the surface-level, but do not work in practice.
>
> To clarify why transitions can’t be relabelled in a batch, each with their own context: the context encoder needs a full batch of transitions to infer a context $z$ - if every transition in the RL batch receives its own hindsight task, it would require computing/inferring a context $z$ for each transition in the batch, therefore requiring an additional batch of “good” context transitions for every RL transition being relabeled, significantly complicating and increasing the time complexity of the sampling procedure.
>
> > *For the experiments; while I understand that the point of the method is to show a step change from the inability to learn with a sparse reward, it’s a little unsettling to see a baseline method that completely flatlines on all tasks. It would be encouraging to see a simple experiment with e.g. a point-mass that needn’t travel as far to achieve a sparse reward (with the expectation that HTR increases sample efficiency, but the baseline clearly works).*
>
> We agree that the poor baseline performance (for the “sparse”/unmodified PEARL) can be a little unsettling or confusing to the reader without further explanation. We will revise Section 4.2 to make it clear that the “sparse” baseline does indeed work on easier sparse reward tasks (assuming the tasks are solvable in a reasonable amount of time through random exploration), and will include a set of experiments on easier environment variations (e.g, PointRobot with a smaller goal radius) to further illustrate this in the paper appendix.

---

> > ### Comment · Reviewer_brT1 · 2021-09-02
> > **Re: Response**
> >
> > I have read the author responses and other reviews. I appreciate the discussion here and trust it can be used to polish the paper somewhat further.
> >
> > Nonetheless, I agree with the other reviewers that the tasks do not particularly highlight the value of the method (to handle settings where dense rewards are difficult to obtain or misleading), or address an obviously difficult new challenge.
> >
> > Overall, I am still inclined to accept the paper but will leave my score unchanged.

---

### Official Review · Reviewer_hvDW · 2021-07-18

**Rating:** 6
**Confidence:** 4

**Summary:**

This paper extends the idea of hindsight experience replay (HER) to a meta reinforcement learning setup. The primary focus is on sparse reward settings, where it is difficult for HER to generalize and even harder to conceptualize and optimize in the meta RL setting.

This paper provides a simple learning algorithm to relabel experiences in an off policy learning settings and provides experimental validation on continuous control tasks.

Key idea: the task is unknown in the meta RL setting. An arbitrary experience trajectory collected during policy evaluation (unknown task) can be reused for an artifically induced task (therefore known). This lets the algorithm leverage past experiences under unknown task distributions to meta learn strategies for an unknown task with sparse rewards.


**Limitations And Societal Impact:**

seems adequate

**Main Review:**


This paper studies an important and interesting problem in meta RL. Given a goal conditioned policy and the ability to explore somewhat efficiently, this algorithm can get the learning signal necessary for the purpose. However, the biggest  limitation is that it won't work for tasks where it is difficult to explore in the first place in order to achieve a positive reward value. So this formulation is deeply tied to exploration feasibfility.

In similar ways, what is the relationship and limitation of this formulation to reachability of goals and reversibility? If certain goal states are irreversible and the episodes are infinite horizon or long range (e.g. in robotics) then what are the failure models?

Does the relabeling probablity K affect results more than in the HER setting?

How does HTR compare to intrinsic motivation based approaches on the same environments (e.g. count or density based)? Is there any inherent advantage for learning to learn over intrinsic motivation? I do think that HTR is an interesting setting by itself to study and explore but it would be helpful to understand and map out how it could go beyond other efficient exploration strategies. Especially because this paper is not claiming to solve tasks that were unsolved before (except under certain assumotion on the reward structure -- which is also entirely reasonable for hypothesis driven research).

I also believe that this setup and algorithm is different HER despite close similarities. This setting is fundamentally studying multiple MDPs which is quite different from HER. I think section 3.1 is quite helpful to understand and appreciate the key differences.

Finally I think the experiments are reasonable and well motivated. The paper could have gotten stronger if there was at least one large scale/real world tasks in the mix, either from robotics, games or navigation. Currently there are simpler experiments in each of these domains.

**Time Spent Reviewing:**

2

---

> ### Author Response · Authors · 2021-08-10
> **Response to Reviewer hvDW**
>
> Thank you for your valuable feedback.
>
> > *Given a goal conditioned policy and the ability to explore somewhat efficiently, this algorithm can get the learning signal necessary for the purpose.*
>
> To clarify a potential misunderstanding - in our work (HTR), the policy is not goal-conditioned, instead it is conditioned on a context variable $z$ (which can potentially encode the goal, but this has to be learned), which itself is inferred from a limited amount of experience in the environment. In HER, the policies are in fact goal-conditioned, i.e., the policy network takes the goal as an explicit input (thus in HER the goal is revealed to the agent, unlike in HTR where the goal, or task, must be inferred from limited experience).
>
> > *However, the biggest limitation is that it won't work for tasks where it is difficult to explore in the first place in order to achieve a positive reward value. So this formulation is deeply tied to exploration feasibfility.*
>
> In the sparse reward environments we study in the paper, it is difficult to explore to achieve a positive reward value (the baseline without relabelling fails to make meaningful progress), however, relabeling with HTR allows the agent to eventually learn an optimal exploration/adaptation strategy. In other words, even if an environment is difficult to explore, as long as the early attempts (e.g., random trajectories) to solve task $T$ can be relabelled as successful attempts for task $T’$ (likely a much easier task than the original $T$), then HTR can be applied.
>
> Instead, the key limitation of our approach is described in Section 3.1, line 222: we assume that this relabelling function (mapping low-reward trajectories to high-reward trajectories) exists, which is a reasonable assumption for goal-reaching environments, but may be much more complex to implement for other environments.
>
> > *In similar ways, what is the relationship and limitation of this formulation to reachability of goals and reversibility?*
>
> Reversibility is not directly related to HTR - whether or not actions leading to a goal are reversible does not impact if or how they can be relabelled and used for training. Similarly, reachability is also a tangential concept to HTR - both the data collected in the replay buffer and the relabelled data will only have positive/non-zero rewards specified by reachable goals.
>
> > *If certain goal states are irreversible and the episodes are infinite horizon or long range (e.g. in robotics) then what are the failure models?*
>
> As mentioned above, the reversibility of goal states does not have any bearing on the specific failure modes of HTR with respect to the base meta-RL algorithm it is built on top of, in our case PEARL. The sparse reward meta-RL setting is already very challenging (as referenced in lines 28-36) - if the task can only be solved after a long sequence of actions, this makes this sparse reward meta-RL setting even more difficult to meta-train in. However, this is also precisely the setting in which a data relabelling approach such at HTR can help - assuming the optimal policy is rewarded near the end of the episode, the agent is less likely to receive sparse reward through random exploration , and therefore may not be able to ever meta-learn an adaptation strategy. In this case, HTR can bootstrap the meta-learning process by training on easier short horizon hindsight tasks.
>
> In the setting where the agent is given an infinite horizon to solve the task (e.g., if the “sparse” PEARL baseline is allowed an infinite number of meta-training steps on PointRobot), then the baseline should be able to eventually solve the task through random exploration (though the amount of time required to do so may be intractable). In this case, HTR will still improve the baseline by increasing its sample efficiency.
>
> > *How does HTR compare to intrinsic motivation based approaches on the same environments (e.g. count or density based)? Is there any inherent advantage for learning to learn over intrinsic motivation? I do think that HTR is an interesting setting by itself to study and explore but it would be helpful to understand and map out how it could go beyond other efficient exploration strategies. Especially because this paper is not claiming to solve tasks that were unsolved before (except under certain assumotion on the reward structure -- which is also entirely reasonable for hypothesis driven research).*
>
> Count or density based methods for encouraging exploration are an interesting point of comparison to HTR and we will revise the Related Work to include additional discussion on this important topic.
>
> Count-based and other similar methods could potentially be used to encourage exploration during meta-training, for example, during the exploration phase of PEARL (where the agent collects a few trajectories to infer the task), an exploration bonus could be applied to encourage the agent to visit a diverse set of states. However, the number of exploration trajectories is typically quite limited (~5 in PEARL), allowing for limited coverage of the state space which reduces the effectiveness of such methods. To counter this, the exploration bonus could be computed across many adaptation attempts, but this would mean the meta-training setup no longer matches the meta-testing setup.
>
> The reason PEARL is unable to make meaningful progress during meta-training (without HTR) is not due to poor exploration. Instead, even with good exploration, the limited number of exploration trajectories means that the probability of receiving positive reward is extremely low, and therefore the agent never receives the positive reward required to train the policy. Therefore it is not clear how effective any count- or density-based method would be in improving PEARL on sparse reward meta-RL environments, especially compared to HTR, which is very effective.
>
> > *Does the relabeling probablity K affect results more than in the HER setting?*
>
> In both HER and HTR, the relabeling probability is an important hyperparameter that should be tuned for optimal performance. With HTR, on our three meta-RL environments, we found that a relabelling probability of 10% ($K = 0.1$) or 30% ($K = 0.3$) was optimal on all three environments. In the original HER paper, on their three goal-conditioned RL environments, the authors found that a relabelling probability of 80% or ~90% worked best on all environments. For both algorithms, there exists a range of relabelling probabilities (also described as a range of ratios of normal-to-relabelled data) where the relabeling algorithm works well, and a range of values where the algorithm performs poorly (the HER authors did not provide results for under 50% relabelled data, which may include more negative results).

---

### Official Review · Reviewer_6Fvi · 2021-08-03

**Rating:** 6
**Confidence:** 4

**Summary:**

This paper proposes an algorithm. Hindsight Task Relabeling, for meta-reinforcement learning in sparse reward environments. It can be viewed as applying the hindsight relabeling in a meta-RL setting, instead of in the goal-conditioned setting of the original HER. With the Meta-RL setting, the task is not revealed to the agent during test time, and thus needs to be inferred as well (compared to the goal-conditioned setting). Specifically, they proposed two relabeling strategies, Single Episode Relabeling (SER) and Episode Clustering (EC). The key difference from naive HER style relabeling is that SER samples an entire episode, subsamples transitions, and then relabels them to the same hindsight task. With EC, similar trajectories are clustered into the same hindsight buffers (e.g. by discretizing the state space into bins) with stratified sampling during model training. The major hyperparameter introduced is K, which determines the probability of sampling original vs hindsight experience when training the model.

The experiments were applied on goal-reaching tasks (2D navigation, wheeled locomotion, and ant locomotion). They used PEARL as the underlying off-policy meta-RL algorithm. The results indicate that HTR is crucial for sparse reward tasks, and can learn as well (sometimes better) than with shaped dense reward. Unlike HER where higher hindsight ratio helped more, in HTR only a K=0.1 (10% hindsight) is best in their environments. Finally, despite EC being more complicated to implement, it performed similarly to the much simpler SER strategy.


**Ethical Concerns:**

No ethical issues with this paper

**Limitations And Societal Impact:**

The broader impacts were discussed in Appendix D. The discussions were fairly general, touching upon the common concerns in deep learning models, as they also apply to the deep meta-RL.

**Main Review:**

**Originality**: The paper can be seen as a generalization (or application) of hindsight relabeling in the goal-conditioned setting to the meta-RL setting. The relabeling strategy needed to be at the episode level in order to mimic the meta-training set up, where they share the same pseudo/real task for the context. The paper discusses the similarity and differences to HER, as well as other sparse reward and unsupervised meta-RL.

**Quality**: The proposed method was evaluated empirically, as there is no theoretical analysis for the proposed framework. However as pointed out in Appendix C4, neither did the original HER paper. Several limitations/assumptions for the method were discussed in section 3.1, including that trajectories can be relabeled under a new task, and that this method could be applied beyond the goal-reaching tasks as proof of concept in this work.

One experiment that would make the need for sparse reward in meta-RL setting convincing is to have an environment where the shaped dense reward (i.e. L2 distance) does not work well. For example, we can have an environment with obstacles, to demonstrate that a hand-engineered dense reward would perform worse than using meta-training on sparse reward with HTR.

**Clarity**: Overall the paper is fairly well written and well organized. One clarification would be the detail about how the hindsight sampling connects with the context vs RL sampler when using PEARL. For example, with SER, how does this interact with the context and RL batches, i.e. is it sampling only 1 episode for the context, and then for the RL batches are sampled independently with the same hindsight task?

**Significance**: The problem of sparse reward in meta-RL settings is a natural extension to existing meta-RL environments. The proposed hindsight task relabeling schemes are relatively straightforward to implement in simple settings where inferring the task is easy/given. It remains to be seen how well this method would scale to more complicated sparse reward tasks beyond goal-reaching type of environments.

**Time Spent Reviewing:**

6

---

> ### Author Response · Authors · 2021-08-10
> **Response to Reviewer 6Fvi**
>
> Thank you for your valuable feedback.
>
> > *One experiment that would make the need for sparse reward in meta-RL setting convincing is to have an environment where the shaped dense reward (i.e. L2 distance) does not work well. For example, we can have an environment with obstacles, to demonstrate that a hand-engineered dense reward would perform worse than using meta-training on sparse reward with HTR.*
>
> We agree that this would be an interesting experiment to further illustrate the differences between HTR and existing approaches for sparse meta-RL (e.g., training PEARL with a separate dense reward function for meta-training). One potential downside of this experiment is it may require careful configuration of the obstacle shape and size to tune the environment difficulty (such that learning to adapt using L2 distance leads to suboptimal behavior compared to learning to adapt on nearby/easily achievable goals first via HTR relabelling, while keeping the environment a reasonable difficulty).
>
> > *One clarification would be the detail about how the hindsight sampling connects with the context vs RL sampler when using PEARL. For example, with SER, how does this interact with the context and RL batches, i.e. is it sampling only 1 episode for the context, and then for the RL batches are sampled independently with the same hindsight task?*
>
> Thank you for raising this question - this is an important implementation detail to discuss in the paper and we will revise Section 3.1 and Appendix C to include the following information:
>
> HTR with SER samples a single episode from the task replay buffer, samples $N$ transitions from that episode and rewrites the rewards for each sampled transition under a new hindsight task. These transitions are used for both the context batch and the RL batch. Alternatively, the context batch and RL batch (of $N$ transitions) can be independently sampled from the same episode, but in practice this implementation detail does not lead to any difference in performance. Either way, HTR with SER is using the same episode (sampled from the original task replay buffer) for both context batches and RL batches.
>
> We found that sampling transitions from the same episode for both the context batch and RL batch is important for performance - the alternative (sampling an episode $e$ from task buffer $b$, choosing a hindsight task $t$, sampling the context batch from $e$ relabelled under $t$, then sampling an RL batch from the entire buffer $b$ relabelled under $t$) often results in low reward in the RL batch, despite having high reward in the context batch. This is because the replay buffer (particularly at the beginning of training) contains a diverse set of trajectories each with a different optimal hindsight task; sampling from the entire task buffer but relabeling under single hindsight task optimal only for a specific trajectory in the buffer can easily lead to a batch of transitions with all zero reward (see Figure 1b).
>
> HTR with EC samples context and RL batches the same way as PEARL: a batch of recent transitions from the buffer is sampled for context, and a batch of transitions from the entire buffer is sampled for the RL batch. The key difference between HTR with EC and PEARL (at the sampling stage) is that a real task buffer is swapped out with a hindsight task buffer according to probability $K$. The other key difference compared to PEARL is at the data collection stage, where relabelled transitions are added to the hindsight task buffers (these two key differences are highlighted in blue in Appendix C, Algorithm 1).

---

> > ### Comment · Reviewer_6Fvi · 2021-08-23
> > **Author Response Acknowledgement**
> >
> > Thank you to the authors for the detailed response to my concerns.
> >
> > On the first point about deceptive dense rewards environments: it is a fair point that designing such environments will be tricky to do, and certainly not feasible within the timeframe of the rebuttal. I think a classic version of U-maze with different length / width of the U may be sufficient as a didactic example, for the future experiments (or follow up work).
> >
> > On the implementation detail: yes it will be great if Section 3.1 and Appendix C includes the discussed information, especially how there were many subtle algorithm design choices in terms of the sampling procedure (i.e. on sampling transitions from the same episode for both the context and RL batch) and the effect on downstream performance. Otherwise future works might not implement it properly and butcher your algorithm when used as a baseline!
> >
> > Overall I am going to keep my original score rating for the paper as a weak accept.

---

### Decision · Program_Chairs · 2021-09-27

**Decision:**

Accept (Poster)

**Comment:**

We thank the authors for rebuttal and the reviewers for engaging in discussions. We reached a consensus for weak acceptance of the paper. However, the method is straightforward given PEARL and HER, and the reviewers and I agree that the current tasks are not challenging enough and do not convincingly demonstrate the value of this approach, and highly encourage the authors to add additional validations.